# Albedo-Induced Global Warming Impact at Multiple Temporal Scales within an Upper Midwest USA Watershed

**Pietro Sciusco** [1,2,*], **Jiquan Chen** [1,2,3], **Vincenzo Giannico** [4], **Michael Abraha** [2,3], **Cheyenne Lei** [1,2,3], **Gabriela Shirkey** [1,2], **Jing Yuan** [2] **and G. Philip Robertson** [3,5,6]

1 Department of Geography, Environment & Spatial Sciences, Michigan State University, East Lansing, MI 48824, USA; jqchen@msu.edu (J.C.); cheyenne@msu.edu (C.L.); shirkeyg@msu.edu (G.S.)
2 Center for Global Change and Earth Observations, Michigan State University, East Lansing, MI 48823, USA; abraha@msu.edu (M.A.); yuanji11@msu.edu (J.Y.)
3 Great Lakes Bioenergy Research Center, Michigan State University, East Lansing, MI 48824, USA; robert30@msu.edu
4 Department of Agricultural and Environmental Sciences, University of Bari A. Moro, 70126 Bari, Italy; vincenzo.giannico@uniba.it
5 W.K. Kellogg Biological Station, Michigan State University, Hickory Corners, MI 49060, USA
6 Department of Plant, Soil, and Microbial Sciences, Michigan State University, East Lansing, MI 48824, USA
* Correspondence: sciuscop@msu.edu; Tel.: +1-(517)-505-1074 or +39-(366)-3820032

**Abstract:** Land surface albedo is a significant regulator of climate. Changes in land use worldwide have greatly reshaped landscapes in the recent decades. Deforestation, agricultural development, and urban expansion alter land surface albedo, each with unique influences on shortwave radiative forcing and global warming impact (GWI). Here, we characterize the changes in landscape albedo-induced GWI ($GWI_{\Delta\alpha}$) at multiple temporal scales, with a special focus on the seasonal and monthly $GWI_{\Delta\alpha}$ over a 19-year period for different land cover types in five ecoregions within a watershed in the upper Midwest USA. The results show that land cover changes from the original forest exhibited a net cooling effect, with contributions of annual $GWI_{\Delta\alpha}$ varying by cover type and ecoregion. Seasonal and monthly variations of the $GWI_{\Delta\alpha}$ showed unique trends over the 19-year period and contributed differently to the total $GWI_{\Delta\alpha}$. Cropland contributed most to cooling the local climate, with seasonal and monthly offsets of 18% and 83%, respectively, of the annual greenhouse gas emissions of maize fields in the same area. Urban areas exhibited both cooling and warming effects. Cropland and urban areas showed significantly different seasonal $GWI_{\Delta\alpha}$ at some ecoregions. The landscape composition of the five ecoregions could cause different net landscape $GWI_{\Delta\alpha}$.

**Keywords:** albedo; global warming impact (GWI); radiative forcing (RF); forest; land conversion; climate regulation; cooling effect; warming effect; landscape composition

## 1. Introduction

Surface albedo—the amount of solar radiation reflected by a surface relative to the total incident solar radiation—is a fundamental component of the Earth's surface energy balance [1,2]. Unlike greenhouse gases (GHGs), which regulate climate by the interception of longwave radiation that affects the Earth's radiation balance, the warming or cooling effects of surface albedo are directly due to instantaneous changes in the amounts of shortwave radiation reflected to outer space. Changes in land use worldwide have greatly reshaped landscapes in recent decades at increasing rates since the Industrial Revolution [3–6]. Earth's surface albedo has changed accordingly, resulting in alterations of the Earth's radiation balance that are partially responsible for the changing climate.

Globally, deforestation, agricultural development—including forest and grassland conversion—and urban expansion are major sources for albedo change [6,7], which, in turn, can directly affect the Earth's radiation balance. Imbalances due to albedo changes are

described by the albedo-induced radiative forcing ($RF_{\Delta\alpha}$, W m$^{-2}$)—changes in the fraction of solar radiation reflected back to the atmosphere from the Earth's surface [8]. For example, according to the Intergovernmental Panel on Climate Change (IPCC; [9]), the RF of well-mixed GHGs has a warming effect equivalent to ~+2.83 W m$^{-2}$, while the $RF_{\Delta\alpha}$ due to land use and land cover change (LULCC) has a cooling effect equivalent to ~−0.15 W m$^{-2}$. In other words, albedo changes have offset ~5% of the energy imbalance caused by well-mixed GHGs, with the offsets varying substantially by region. However, this offset is a global average in reference to LULCC, with dominant changes from forest to non-forest since 1750. Therefore, the local contributions of $RF_{\Delta\alpha}$ due to LULCC are unknown and might play an important role in the overall global average of climate regulation effects.

The present scientific understanding of the forcing effects of albedo due to LULCC is ranked as medium-low relative to the rich scientific evidence of the forcing effects of GHGs [1]. To cross-examine effects with the climate impact of other GHGs (i.e., biogeochemical GWI), albedo-induced warming or cooling can be converted into equivalents of carbon-dioxide ($CO_{2eq}$) and/or carbon ($C_{eq}$) atmospheric radiative forcing via the concept of albedo-induced global warming potential ($GWP_{\Delta\alpha}$, kg $CO_{2eq}$ m$^{-2}$ yr$^{-1}$; [9,10]) metric—hereinafter referred to as global warming impact ($GWI_{\Delta\alpha}$), to be in line with our previous study [11]. For example, Houspanossian et al. [12] found that conversion from forests to croplands, from forests to pastures, and from pastures to croplands in dry subtropical forests of South America offset 12–27 Mg $C_{eq}$ ha$^{-1}$ during a 12-year period, or from 15% to 55% of the total C emissions due to deforestation. In Europe, Carrer et al. [13] reported that inclusion of cover crops in annual cropping systems could have cooling effects equivalent to a mitigation of −0.03 Mg $C_{eq}$ ha$^{-1}$ yr$^{-1}$, while Lugato et al. [14] showed that such mitigation potential due to the inclusion of cover crops could be substantially enhanced by growing high-albedo chlorophyll-deficient cover crops. In southwest Michigan, USA, Chen et al. [15] estimated that land conversion from forest to maize (*Zea mays* L.) can provide a cooling equivalent to a mitigation of −0.043 Mg $C_{eq}$ ha$^{-1}$ yr$^{-1}$ due to a 0.051 (i.e., 5.1%) increase in albedo. At watershed scale, Sciusco et al. [11] demonstrated that altered landscapes could produce cooling effects relative to the intact, native, late successional forests typical of pre-European settlement and contribute a range of −0.1 to −0.4 Mg $C_{eq}$ ha$^{-1}$ yr$^{-1}$, which is the same order of magnitude of biogeochemical GWI emissions due to many crop management components [16,17].

Despite the potential importance of albedo modification strategies for regional to global climate mitigation by IPCC [18] and numerous discussions [19,20] on intentionally increasing surface albedo to cool the Earth, little effort has been made to understand changes in landscape $RF_{\Delta\alpha}$ or $GWI_{\Delta\alpha}$ in the context of landscape composition at broader temporal scales and for multiple anthropogenic LULCC [11,21,22]. A critical unknown is how different cover types contribute to total landscape $GWI_{\Delta\alpha}$ at different times of the year. Over the long term (i.e., years to decades), little is known about whether the intra-annual variations of landscape $GWI_{\Delta\alpha}$ are significant.

Here, we build on Sciusco et al. [11] to estimate the contributions of the $GWI_{\Delta\alpha}$ to the landscape warming or cooling effects at seasonal and monthly timescales over 19 years for multiple ecoregion subtypes. We quantify different cover types in different ecoregions of the upper Midwest USA watershed by estimating (1) the monthly and seasonal contributions to the total landscape cooling or warming; (2) the variations of $GWI_{\Delta\alpha}$ contributions by cover type, ecoregion, and year; and (3) the magnitude of cooling or warming effects due to land cover change relative to mature forest cover.

## 2. Materials and Methods

### 2.1. Study Area and Landscape Composition

The Kalamazoo River Watershed (5621 km$^2$; Figure 1) is located in southwest Michigan, USA, and includes portions of 10 counties: Allegan, Barry, Calhoun, Eaton, Hillsdale, Jackson, Kalamazoo, Kent, Ottawa, and Van Buren. The mean annual temperature (1981–2010) is 9.9 °C, and the average annual precipitation is 900 mm evenly distributed throughout

the year [23]. The dominant cover type prior to European settlement in the early 1800 s was eastern broadleaf deciduous forest [24], with scattered patches of tallgrass prairie, oak savanna, lakes, and wetlands [25]. The dominant land cover includes cultivated crops, successional forest stands, pasture-hay grasslands, and two urban areas (i.e., Kalamazoo and Battle Creek). Medium to coarse texture soils and mesic climate allow the continuous recharge of groundwater [26].

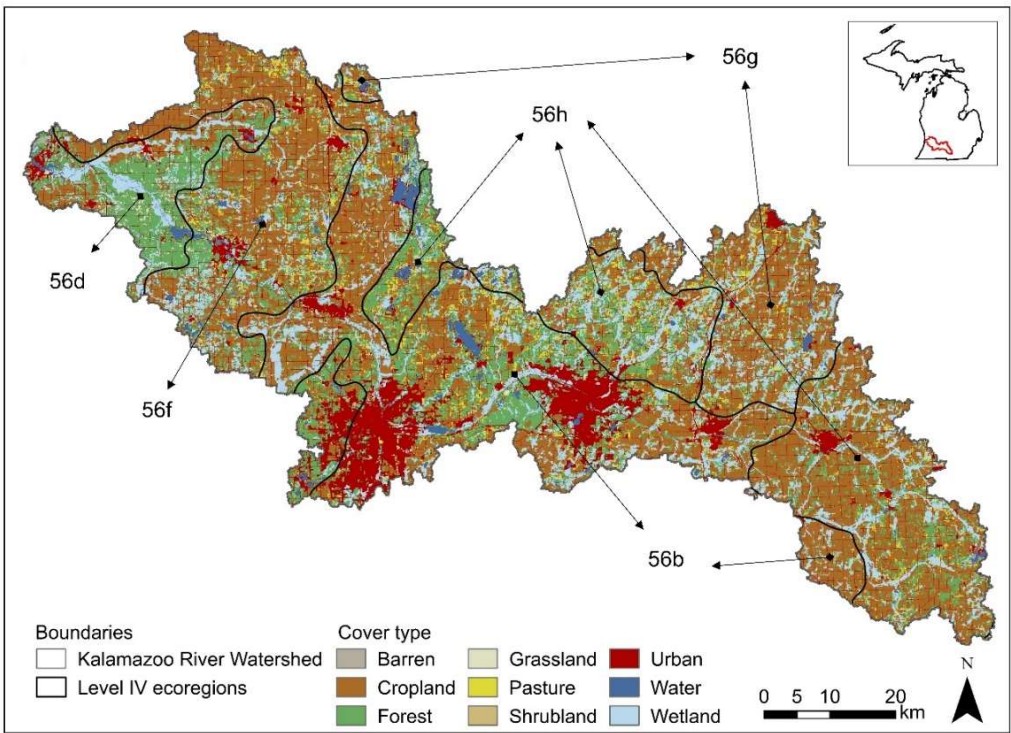

**Figure 1.** The study area of the five United States Environmental Protection Agency Level IV ecoregions and the nine National Land Cover Database cover type classes within the Kalamazoo River Watershed in 2001. (Map projection: WGS-84 UTM Zone 16 N).

Within the watershed, there are five United States Environmental Protection Agency (US EPA) Level IV ecoregions (Figure 1). Level IV ecoregions have the finest resolution and exhibit unique physiographic, geologic, pedologic, botanic, hydrologic, and climatic characteristics [27]. The five ecoregions studied here are Battle Creek Outwash Plain (56b), Michigan Lake Plain (56d), Lake Michigan Moraines (56f), Lansing Loamy Plain (56g), and Interlobate Dead Ice Moraines (56h). For further details, see the US EPA [28].

In this study, we used the National Land Cover Database (NLCD; [29,30]), which provides nine land cover types (barren, cropland, forest, grassland, pasture, shrubland, urban, water, and wetland) at $30 \times 30$ m spatial resolution, with an overall accuracy ranging between 80% and 90%, in the central and western U.S. [29]. Land cover classifications from NLCD are available for the years 2001, 2004, 2006, 2008, 2011, 2013, and 2016. Because land cover maps from NLCD are not annually provided like albedo data (see next section), we assumed land cover is similar to the prior year for years where NLCD data are not available. For example, the land cover map for 2002 was assumed to be the same as that of 2001, and for 2017–2019, we assumed the land cover had no significant changes.

## 2.2. Intra-Annual Changes in Albedo

We obtained instantaneous albedo data at 10:30 a.m. local time (MODIS Terra morning overpassing time) at $500 \times 500$ m spatial resolution and at daily time-step for 2001–2019 from the most recent collection (V006) of the MODIS Bidirectional Reflectance Distribution Function (BRDF) MCD43A3 product [31]. The MCD43A3 product contains both black-sky

(i.e., directional-hemispherical reflectance) and white-sky (i.e., bi-hemispherical reflectance) albedos [32]. We considered white-sky albedo ($\alpha$) at a shortwave length of 0.3–5.0 μm by growing season and month during 2001–2019.

Growing season and monthly albedos ($\alpha_{gs}$ and $\alpha_{mo}$, respectively) at 10:30 a.m. local time were derived by stacking (i.e., median image composite) the daily images into growing seasons or months by year. Specifically, $\alpha_{gs}$ accounted for 19 composites (2001–2019), while $\alpha_{mo}$ accounted for 11 composites (January–December, over the 19-year period, less March, which we removed because few images were available and likely due to high cloud cover). We applied the same methodology as Jeong et al. [33] and Sciusco et al. [11] to identify the growing season (roughly from March to November) for each year. Briefly, we used the enhanced vegetation index (EVI; see Appendix A) to identify the growing season by detecting the EVI inflection points (i.e., the dates) when maximum and minimum change rate in greenness occurred over the entire watershed (which was roughly between March and November for the 19-year period).

We employed the Google Earth Engine platform [34] to analyze and process all datasets (see Appendix A). We then performed a zonal statistical analysis within ArcMap (v. 10.6) to calculate the proportion of NLCD cover types within each MODIS pixel, and to extract $\alpha_{gs}$ and $\alpha_{mo}$ values by pixel before statistical analysis in RStudio v.1.2.5033 [35].

*2.3. Albedo-Induced Global Warming Impact (GWI$_{\Delta\alpha}$)*

We employed the linear downscaling approach of Chen et al. [36] to estimate surface albedo of cover type $i$ ($\hat{\alpha}_{si}$) at each MODIS pixel. For a MODIS pixel, $\hat{\alpha}_{si}$ is considered as the sum of surface albedo of cover type $i$ ($\alpha_{si}$) within each MODIS pixel, as follows [36]:

$$\hat{\alpha}_{si}(t) = \left[ \sum (k_i \times \alpha_{si}(t)) \right] + \varepsilon_t \tag{1}$$

where $\hat{\alpha}_{si}$ is the estimated surface albedo of cover type $i$ for a time-period ($t$) (i.e., GS: $t$ = 2001–2019 and monthly: $t$ = January–December, less March), $k_i$ is the proportion (0–1) of cover type $i$ in each MODIS pixel, and $\varepsilon_t$ represents model residuals.

The calculation of landscape albedo-induced radiative forcing (RF$_{\Delta\alpha}$) and global warming impact (GWI$_{\Delta\alpha}$) is based on the change in surface albedo due to land cover conversion. This is normally considered as the surface albedo difference ($\Delta\alpha_s$) between a cover type $i$ and the native vegetation cover type (i.e., the reference) [11]. Here, forest is the dominant land cover type prior to European settlement and serves as our reference [24]. Thus, the surface albedo difference of cover type $i$ ($\Delta\alpha_{si}$) for growing season and monthly periods and for each ecoregion at 10:30 a.m. local time is calculated as:

$$\Delta\alpha_{si} = (\hat{\alpha}_{si} - \hat{\alpha}_{sf}) \tag{2}$$

where $\hat{\alpha}_{si}$ and $\hat{\alpha}_{sf}$ are the estimated surface albedos of cover type $i$ and of the reference forest $f$ for growing season and monthly periods and for each ecoregion. We calculated $\Delta\alpha_{si}$ only when the proportion of a cover type $i$ was ≥80% of the 500 × 500 m MODIS pixel. Conversions to barren, grassland, and shrubland were excluded, as they have a small percentage of the total study area, as were current water and wetland covers. Consequently, only cropland, forest, pasture, and urban covers were considered in this study.

We then used $\Delta\alpha_{si}$ to calculate the landscape RF$_{\Delta\alpha}$ as follows [13,37,38]:

$$RF_{\Delta\alpha} = -(SW_{in} \cdot K_T \cdot \Delta\alpha_{si}) \tag{3}$$

where $RF_{\Delta\alpha}$ (W m$^{-2}$) is the landscape albedo-induced radiative forcing at the top-of-atmosphere at 10:30 a.m. local time, and $SW_{in}$, $K_T$, and $\Delta\alpha_{si}$ are the incident shortwave radiation at the surface, the clearness index (for more details, see Equations (A1)–(A5) in Appendix A), and the surface albedo difference between a cover type $i$ and the reference forest, respectively, for the growing season and monthly periods and for each ecoregion (Equation (2)). The incident shortwave radiation at the surface ($SW_{in}$) and the clearness

index ($K_T$) were derived from the solar and meteorological dataset NASA POWER [39] at daily time-step for multiple locations (i.e., five Level IV ecoregions) within the Kalamazoo River Watershed. We then averaged $SW_{in}$ and $K_T$ values to match the 19 growing seasons and the 11 months. Positive or negative values of $RF_{\Delta\alpha}$ indicate warming and cooling effects, respectively.

Lastly, we calculated the landscape $GWI_{\Delta\alpha}$ as follows [13,37,38]:

$$GWI_{\Delta\alpha} = \left( \frac{A \cdot RF_{\Delta\alpha}}{AF(t) \cdot rf_{CO_2}} \cdot \frac{1}{TH} \right) \qquad (4)$$

where $GWI_{\Delta\alpha}$ (kg $CO_{2eq}$ m$^{-2}$ yr$^{-1}$) is the landscape albedo-induced global warming impact for the growing season and monthly periods in each ecoregion at 10:30 a.m. local time, $A$ is the area for which the hypothesized albedo change occurred (here normalized to 1 m$^2$), $AF(t)$ is the $CO_2$ airborne fraction that remains in the atmosphere at time *(t)* following a single pulse emission, $rf_{CO_2}$ is the marginal RF for $CO_2$ emissions at a given atmospheric concentration, and $TH$ represents the time horizon of global warming. The parameter $AF(t)$ is modeled with an exponential function through a multi-model impulse response function analysis (for more details, see Equation (A6) in Appendix A) [40], while $rf_{CO_2}$ is kept constant at 0.908 W kg $CO_2^{-1}$ [13,41,42] and $TH$ is fixed at 100 years (i.e., the number of time steps the GWI is then divided by) [43,44]. With Equation (4), we calculate the equivalent $RF_{\Delta\alpha}$ that a unit area of $A$ would have at the global scale. Positive or negative values of the $GWI_{\Delta\alpha}$ indicate effects equivalent to $CO_2$ emission or mitigation, respectively. Here, we report the results of landscape seasonal and monthly $GWI_{\Delta\alpha}$ ($GWI_{\Delta\alpha gs}$ and $GWI_{\Delta\alpha mo}$, respectively) expressed with units of Mg C$_{eq}$ ha$^{-1}$ gs$^{-1}$ and Mg C$_{eq}$ ha$^{-1}$ mo$^{-1}$, respectively, which refer to 10:30 a.m. local time.

### 2.4. Contributions of Land Cover Change to $GWI_{\Delta\alpha}$

We performed a nested analysis of variance (ANOVA) with repeated measurements (i.e., growing season and monthly periods) to quantify the contribution to landscape $GWI_{\Delta\alpha}$ by cover type and ecoregion for the growing season and monthly periods. Specifically, we looked at contributions among and within the five ecoregions, with two linear models:

$$GWI_{\Delta\alpha}(t) = (ecoregion \times cover\ type)(t) \qquad (5)$$

$$GWI_{\Delta\alpha}(t) = cover\ type(t) \qquad (6)$$

where *ecoregion* refers to the five Level IV ecoregions (i.e., 56b, 56d, 56f, 56g, and 56h), and *cover type* refers to the three cover types (i.e., cropland, pasture, and urban) used to determine the albedo difference from the reference forest (Equation (2)) for growing season and monthly periods at each ecoregion. For further details about the ANOVA analysis, see Appendix A.

## 3. Results

### 3.1. Land Use and Land Cover Change

The dominant land cover types in the Kalamazoo River Watershed—cropland, forest, pasture, and urban cover—underwent detectable changes during 2001–2016 (Table 1). Declines were the highest for forest (1500 ha, −1.3%), followed by pasture (1087 ha, −3.83%) and cropland (714 ha, −0.33%).

Gains occurred for urban cover type (~2000 ha, ~+3%). Cropland, forest, and pasture were converted into a variety of cover types (Table 2): Cropland was primarily converted into urban (1157 ha), pasture (776 ha), and forest (137 ha), forest was largely converted into urban (515 ha) and cropland (164 ha), while pasture was mostly converted into cropland (1456 ha) and urban (197 ha).

**Table 1.** Land cover composition in ha (%) of the Kalamazoo River Watershed, and net gain (+) and loss (−) for each cover type during the period 2001–2016.

| Cover Type | Land Cover Composition | | Land Cover Gain (+) or Loss (−) |
|---|---|---|---|
| | ha (%) | | |
| | 2001 | 2016 | 2001−2016 |
| Cropland | 215,869 (40.90) | 215,155 (40.80) | −714 (−0.33) |
| Forest | 115,393 (21.90) | 113,893 (21.60) | −1500 (−1.30) |
| Pasture | 28,382 (5.40) | 27.296 (5.20) | −1087 (−3.83) |
| Urban | 66,759 (12.70) | 68,743 (13.00) | +1984 (+2.89) |

**Table 2.** Pivot table showing the land cover conversion (ha) of each cover type during the period 2001–2016 across the Kalamazoo River Watershed. Bold values indicate the main land cover conversions, while "*" indicates no land cover conversion.

| 2001 \ 2016 | Land Cover Conversion (ha) 2001–2016 | | | | |
|---|---|---|---|---|---|
| | Cropland | Forest | Pasture | Urban | Total$_{2001}$ |
| Cropland | 212,802 * | **137** | **776** | **1157** | 214,872 |
| Forest | **164** | 113,337 * | 9 | **515** | 114,025 |
| Pasture | **1456** | 57 | 26,467 * | **197** | 28,176 |
| Urban | 16 | 21 | 8 | 66,704 * | 66,748 |
| Total$_{2016}$ | 21,4437 | 113,552 | 27,259 | 68,573 | 423,821 |

*3.2. Albedo and GWI$_{\Delta\alpha}$ in Time and Space*

The linear downscaling model (Equation (1)) showed that each cover type contributed differently to the total $\alpha_{gs}$ (adj. $R^2$ = 0.995) and $\alpha_{mo}$ (adj. $R^2$ = 0.745) (Table S1). The four cover types had an overall average $\alpha_{gs}$ of 0.16 ± 0.013 (Table S2), with cropland having the highest $\alpha_{gs}$ at 0.17 ± 0.002 (Table S2), followed closely by pasture (0.16 ± 0.003), urban (0.15 ± 0.002), and forest (0.15 ± 0.004) covers. On the other hand, the $\alpha_{mo}$ (overall average: 0.23 ± 0.134) had a higher variation than $\alpha_{gs}$ and was higher in January, February, and December, with a maximum of 0.46 ± 0.109 in February (Table S2). Other months exhibited lower $\alpha_{mo}$, with a minimum of 0.14 ± 0.017 in November. Cropland and pasture had the highest $\alpha_{mo}$ (0.28 ± 0.183 and 0.25 ± 0.156, respectively). However, the remaining cover types ranked differently for $\alpha_{mo}$ than for $\alpha_{gs}$, decreasing in the order urban (0.22 ± 0.119) and forest (0.19 ± 0.080). As with $\alpha$, $\Delta\alpha$ varied substantially across cover types but was relatively constant within a cover type among the ecoregions (Figure 2a,b). In particular, $\Delta\alpha_{gs}$ ranged from −0.001 ± 0.003 for urban to 0.026 ± 0.004 for cropland cover (Figure 2a), while $\Delta\alpha_{mo}$ had higher variation and ranged from 0.024 ± 0.040 for urban to 0.089 ± 0.108 for cropland cover (Figure 2b).

The average GWI$_{\Delta\alpha}$ showed an overall cooling effect (Figure 3a,b) for most cover types, with the exception of urban, which showed neutral effects. Overall, cropland cover type had seasonal and monthly average cooling effects equivalent to −0.35 ± 0.05 Mg C$_{eq}$ ha$^{-1}$ gs$^{-1}$ and −0.68 ± 0.61 Mg C$_{eq}$ ha$^{-1}$ mo$^{-1}$, respectively (Figure 3a,b). The highest seasonal and monthly cooling effects reached −0.47 Mg C$_{eq}$ ha$^{-1}$ gs$^{-1}$ (in 2015 for Ecoregions 56b, 56d, 56f, and 56h; Tables S3.1–3.3, and 3.5 and −2.15 Mg C$_{eq}$ ha$^{-1}$ mo$^{-1}$ (in February for Ecoregion 56b; Table S4.1), for the two periods, respectively. These cooling effects represented ~26% and ~68% more than the seasonal and monthly annual averages, respectively. On the other hand, urban cover was the only cover type showing neutral effects (i.e., seasonal and monthly average effects equivalent to −0.001 ± 0.034 Mg C$_{eq}$ ha$^{-1}$ gs$^{-1}$ and −0.158 ± 0.256 Mg C$_{eq}$ ha$^{-1}$ mo$^{-1}$, respectively; Figure 3a,b).

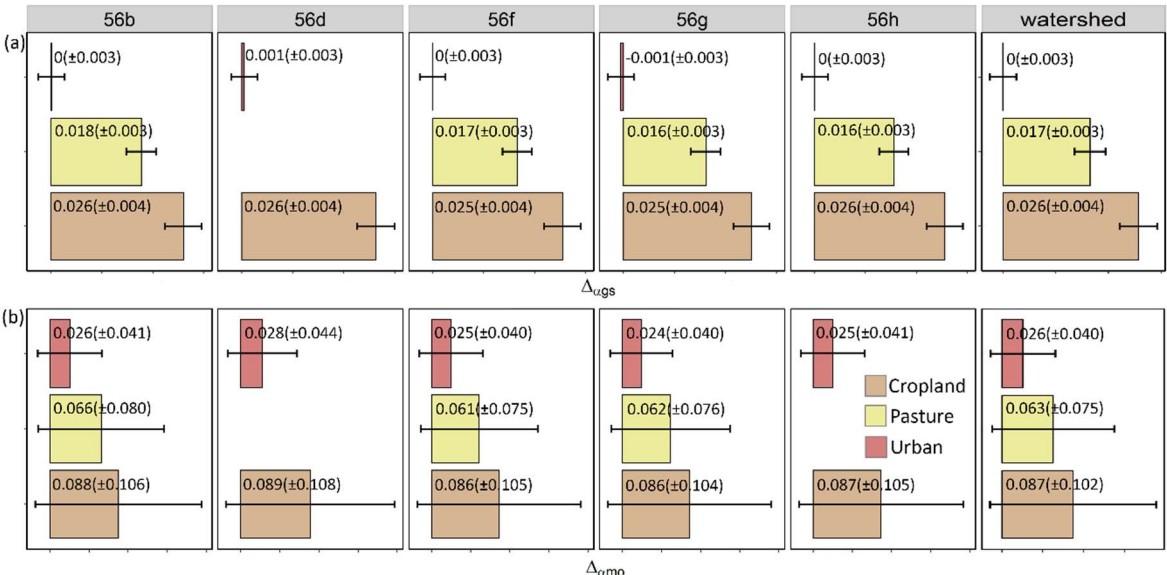

**Figure 2.** Albedo difference ($\Delta\alpha$) between a cover type *i* and the forest for (**a**) growing season ($\Delta\alpha_{gs}$) and (**b**) monthly ($\Delta\alpha_{mo}$) periods, within the five Level IV ecoregions and the entire Kalamazoo River Watershed 2001–2019. Annotations indicate mean ($\pm$standard deviation) of $\Delta\alpha_{gs}$ and $\Delta\alpha_{mo}$.

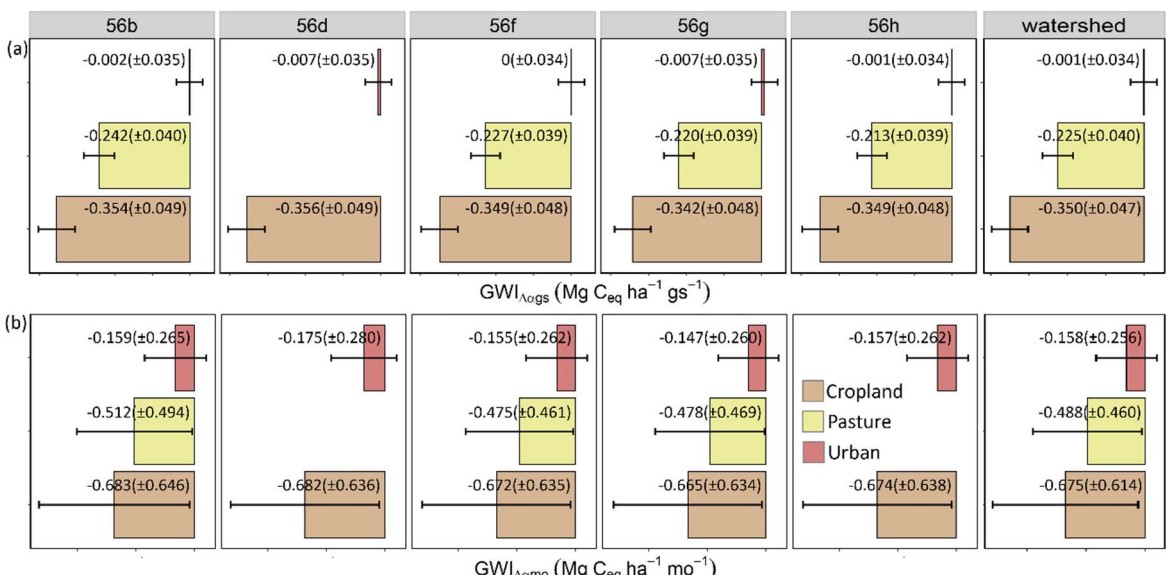

**Figure 3.** Albedo-induced global warming impact ($GWI_{\Delta\alpha}$) for (**a**) growing season ($GWI_{\Delta\alpha gs}$) and (**b**) monthly ($GWI_{\Delta\alpha mo}$) periods, within the five Level IV ecoregions and the entire Kalamazoo River Watershed 2001–2019. Annotations indicate mean ($\pm$standard deviation) of $GWI_{\Delta\alpha gs}$ and $GWI_{\Delta\alpha mo}$.

All three cover types showed similar trends in $GWI_{\Delta\alpha gs}$ 2001–2019 across the five ecoregions (Tables S3.1–3.5 and Figure 4a) with similar deviations from the $GWI_{\Delta\alpha gs}$ mean. The growing season cooling effects ($GWI_{\Delta\alpha gs}$) ranged between $-0.39$ and $-0.47$ Mg $C_{eq}$ $ha^{-1}$ $gs^{-1}$ for cropland, between $-0.25$ and $-0.31$ Mg $C_{eq}$ $ha^{-1}$ $gs^{-1}$ for pasture, and between $-0.01$ and $-0.11$ Mg $C_{eq}$ $ha^{-1}$ $gs^{-1}$ for urban. Urban was the only cover type that also showed warming effects, with $GWI_{\Delta\alpha gs}$ raging between 0.01 and 0.05 Mg $C_{eq}$ $ha^{-1}$ $gs^{-1}$. The inter-monthly variation of $GWI_{\Delta\alpha mo}$ for cropland, pasture, and urban cover showed similar trends with higher cooling effects in January, February, and December (Tables S4.1–4.5 and Figure 4b). Among the three cover types, the highest cooling occurred in cropland ($-0.18$ to $-2.15$ Mg $C_{eq}$ $ha^{-1}$ $mo^{-1}$), followed by pasture ($-0.13$ to $-1.64$ Mg $C_{eq}$ $ha^{-1}$ $mo^{-1}$) and urban ($-0.02$ to $-0.470$ Mg $C_{eq}$ $ha^{-1}$ $mo^{-1}$) cover. For the

three cover types, $GWI_{\Delta\alpha mo}$ was relatively constant from April to November. However, urban had slightly bell-shaped trends with small warming effects in June and July (0.01 to 0.1 Mg $C_{eq}$ $ha^{-1}$ $mo^{-1}$; Tables S4.1–4.5 and Figure 4b), and cropland had an inverted bell-shaped trend with slight rises in June and October. Lastly, pasture (in Ecoregions 56b, 56f, and 56g) had relatively constant trends. The variation among ecoregions in $GWI_{\Delta\alpha gs}$ was significant ($p < 0.001$) by ecoregion, cover type, and their interactions (ANOVA model in Equation (5); Table S5), while the variation in $GWI_{\Delta\alpha mo}$ was significant ($p < 0.001$) only by cover type. Neither years nor months were significant for $GWI_{\Delta\alpha}$ variations. Most of the variation in $GWI_{\Delta\alpha gs}$ was explained by cover type ($\eta^2 = {\sim}95\%$), followed by the interaction between ecoregions and cover type ($\eta^2 = {\sim}52\%$) and ecoregions ($\eta^2 = 32\%$). In comparison, the variation in $GWI_{\Delta\alpha mo}$ was almost equally explained by ecoregion, cover type, and their interactions, although only cover type was significant ($\eta^2 = {\sim}24\%$ at $p < 0.001$). The variation in both $GWI_{\Delta\alpha gs}$ and $GWI_{\Delta\alpha mo}$ within ecoregions (Equation (6); Table S5), however, was significant ($p < 0.001$) by cover type, which explained more of the variation in $GWI_{\Delta\alpha gs}$ ($\eta^2 = 99\%$) than in $GWI_{\Delta\alpha mo}$ ($\eta^2 = 65\%$).

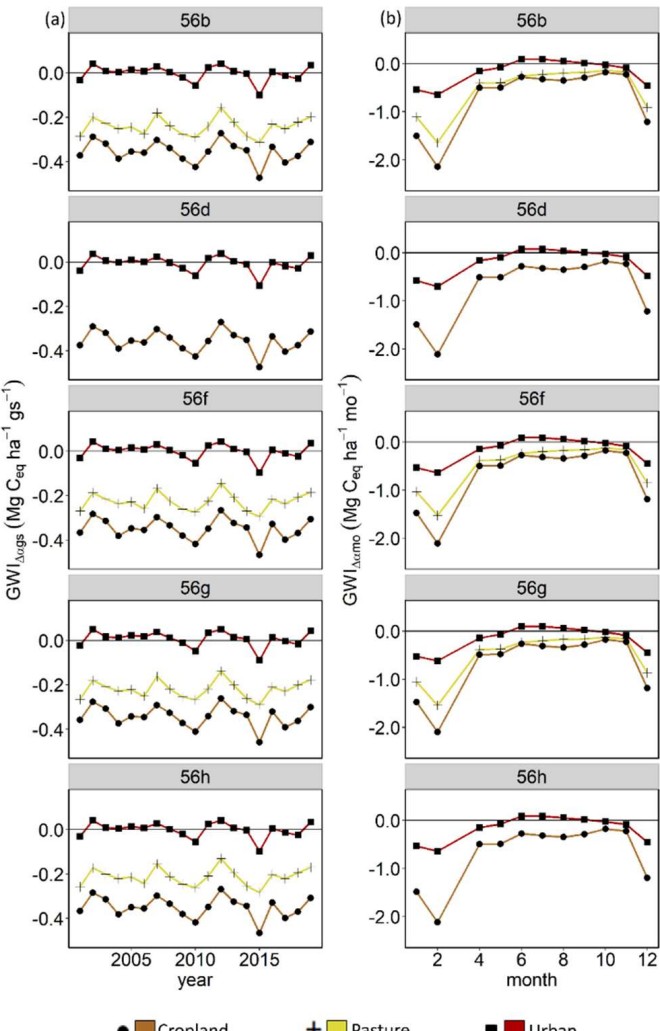

**Figure 4.** Albedo-induced global warming impact ($GWI_{\Delta\alpha}$) for a given cover type within the five Level IV ecoregions. Panels (**a**,**b**) represent the $GWI_{\Delta\alpha}$ for the growing season ($GWI_{\Delta\alpha gs}$; 2001–2019) and monthly ($GWI_{\Delta\alpha mo}$; January–December, less March) periods, respectively. Positive and negative values of $GWI_{\Delta\alpha}$ indicate warming and cooling effects, respectively, equivalent to Carbon ($C_{eq}$) emission and mitigation, respectively.

A post-hoc Tukey test analysis (Figure 5a,b) showed that, within each ecoregion, the least square means (LSMs) of $GWI_{\Delta\alpha gs}$ had low variability and were significantly different among the cover types (Figure 5a). The LSMs for $GWI_{\Delta\alpha mo}$ were more variable, and many cover types had statistically similar means (Figure 5b). Among ecoregions, the LSMs of cropland $GWI_{\Delta\alpha gs}$ at Ecoregion 56g were significantly different from those at Ecoregions 56b and 56d, while the LSMs at urban $GWI_{\Delta\alpha gs}$ at Ecoregion 56g were significantly different from those at Ecoregion 56d. However, no significant differences in their LSMs for $GWI_{\Delta\alpha mo}$ were observed (Figure 5b).

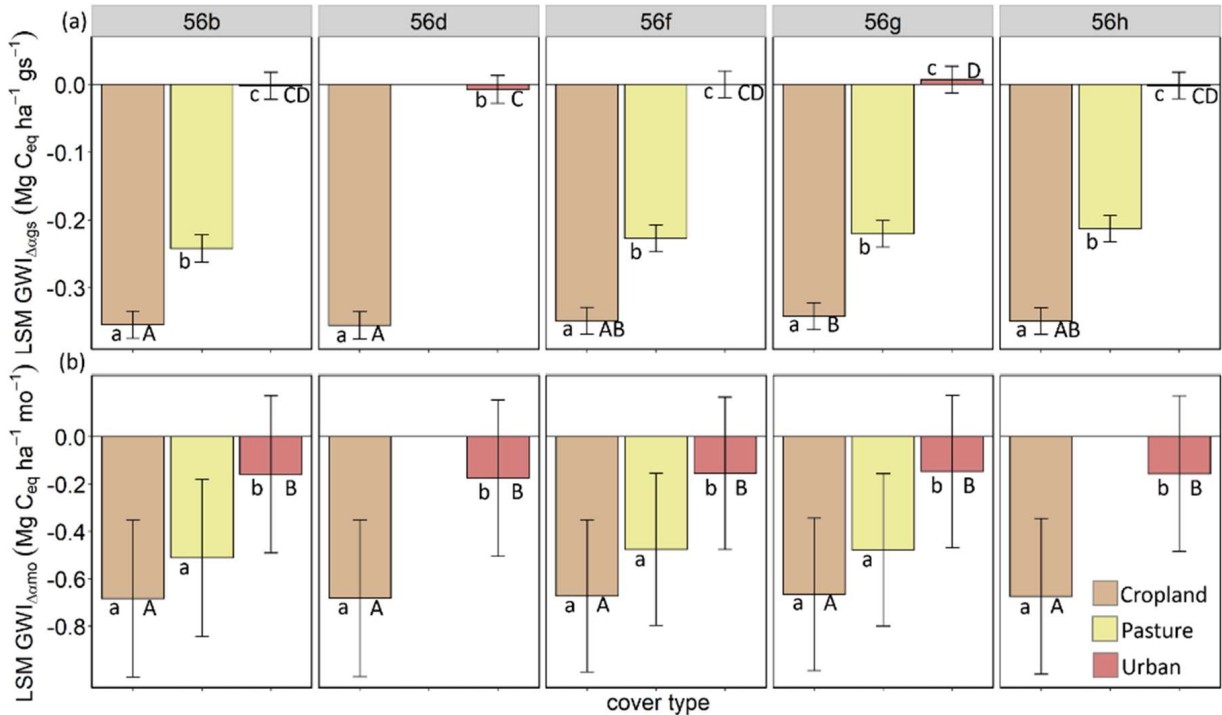

**Figure 5.** The least square means (LSMs) multi-comparison analysis of albedo-induced global warming impact ($GWI_{\Delta\alpha}$) for a given cover type across the five Level IV ecoregions for the (**a**) growing season ($GWI_{\Delta\alpha gs}$; 2001–2019) and (**b**) monthly ($GWI_{\Delta\alpha mo}$; January–December) periods. Whiskers represent the lower and upper limits of the 95% family-wise confidence level of the LSMs. Bars sharing the same letters are not significantly different according to the post-hoc Tukey test analysis. Lowercase letters indicate differences among cover types within the five ecoregions, while uppercase letters indicate differences of same cover type among the five ecoregions. The among ecoregions analysis only considered cropland and urban covers (i.e., the cover types that were in every ecoregion).

The overall $GWI_{\Delta\alpha}$ contribution from different seasons and months varied by cover type, and it was exclusively higher during the non-growing season (NGS) than during the growing season (GS) months for all ecoregions (Table S6 and Figure 6), with the NGS months being characterized by only cooling effects (Table S7). As a general trend, the highest contributions were in February and the lowest in October. During the NGS, urban (at all ecoregions) contributed the most to the total cooling effect (between 18% and 31%), followed by pasture (at Ecoregions 56b, 56f, and 56g; contribution between 14% and 25%), and cropland (at all ecoregions; contribution between 14% and 24%). It is worth noting that during GS months, no cover type had a contribution >8% (i.e., 1/12 of the annual total). Nevertheless, climate regulations of urban (for all ecoregions) were close to the overall mean value of 8% in April, while cropland (at all ecoregions) and pasture (at Ecoregions 56b, 56f, and 56g) were close in April and May.

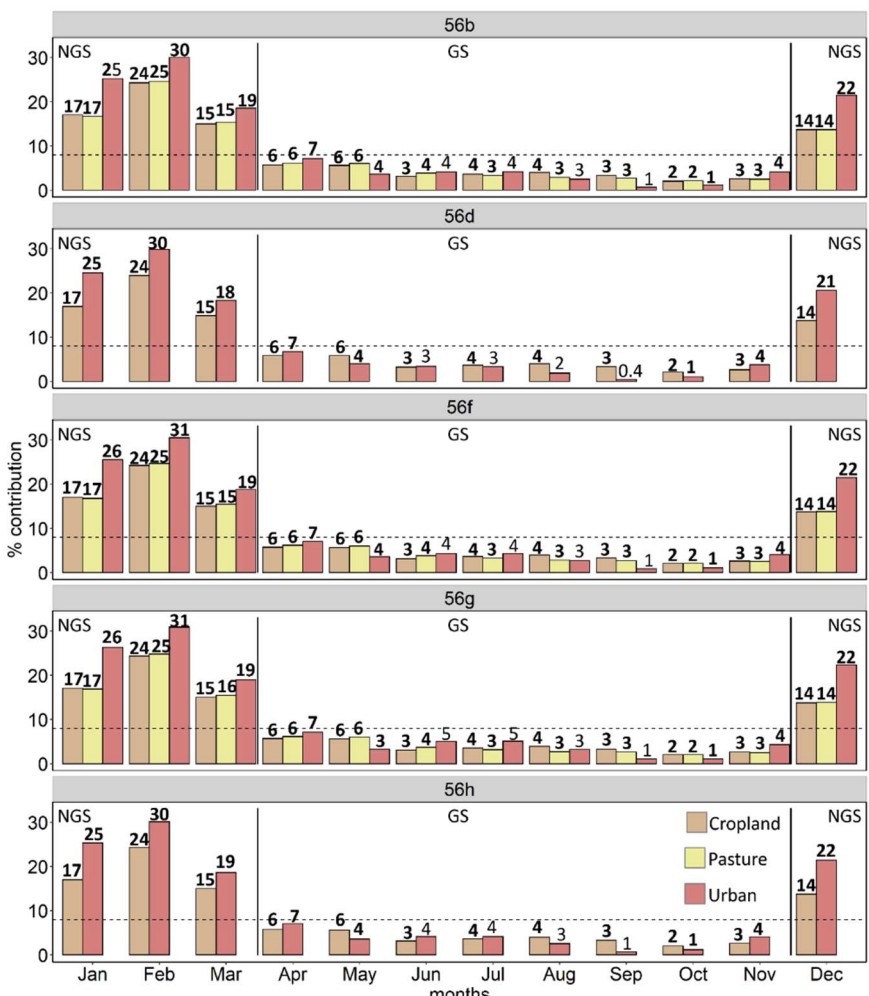

**Figure 6.** Percent contribution of albedo-induced global warming impact (GWI$_{\Delta\alpha}$) to cooling (values in bold) or warming effects by season and month periods for major cover types in the five Level IV ecoregions. The horizontal dashed line represents the average contribution ~8% (i.e., 1/12 of the annual total), and the solid vertical lines separate the non-growing season (NGS) from the growing season (GS) months. Values of March were missing and gap-filled as the mean GWI$_{\Delta\alpha}$ of February and April.

## 4. Discussion

Our results showed that the albedo-induced global warming impact (GWI$_{\Delta\alpha}$) accounted for a significant portion of the climate cooling effects (i.e., C$_{eq}$ mitigation) due to land cover changes and landscape composition. Individual contributions varied by cover type, ecoregion, and season/month, with cropland showing the highest cooling effects, followed by pasture. Urban showed both cooling and warming effects, the latter of which occurred only during growing season months. Overall, the cooling effects of the monthly GWI$_{\Delta\alpha}$ were higher than the seasonal ones, most likely due to the substantial influence of snow cover on land surface albedo (i.e., high presence of snow during the winter months; [45]) combined with the effect of management practices on different vegetation surfaces. For example, snowfalls on harvested crop fields create a highly reflective layer, whereas on forested fields, snowfalls tend to be masked by the tree structures and reflect less. For the same reason, seasonal analysis showed that the cooling contributions during the non-growing season months were higher than during the growing season months. Overall, our results seem to be promising in the context of climate regulation of albedo changes due to land cover changes and landscape composition. Nevertheless, several assumptions and limitations to our study could benefit GWI computations elsewhere.

### 4.1. Cooling Effects

During the 19-year study period, the highest cropland cooling effect of the seasonal and monthly $GWI_{\Delta\alpha}$ was equivalent to $-0.47$ Mg $C_{eq}$ ha$^{-1}$ gs$^{-1}$ in 2015 for Ecoregions 56b, 56d, 56f, and 56h, and $-2.15$ Mg $C_{eq}$ ha$^{-1}$ mo$^{-1}$ in February for Ecoregion 56b, respectively. In comparison, Abraha et al. [46], accounting for GHGs using whole-system lifecycle analysis, found emissions of 2.6 Mg $C_{eq}$ ha$^{-1}$ yr$^{-1}$ over 8 years in Conservation Reserve Program (CRP) grasslands converted to maize. Thus, the seasonal and monthly maximum albedo-induced cooling effects from cropland represent (i.e., offset) 18% and ~83%, respectively, of the annual $C_{eq}$ over the 8 years from CRP grasslands converted to maize fields. The total highest albedo cooling effect due to both the seasonal and monthly $GWI_{\Delta\alpha}$ would completely offset the annual emissions due to the grassland converted to maize over the 8 years. Moreover, the abovementioned cropland cooling effects are more than enough to offset the annual net biogeochemical GWI—i.e., warming effects due to the net contributions from $CO_2$, methane ($CH_4$), and nitrous-oxide ($N_2O$) at 0.31 Mg $C_{eq}$ ha$^{-1}$ yr$^{-1}$—produced by annual crop systems (i.e., maize-soybean-wheat rotation) under conventional tillage management of the same area [47].

### 4.2. Variable Effects

Urban areas appeared to have either cooling or warming effects depending on the temporal scale examined. Unlike cropland, which had cooling effects during summer months mostly due to changes in vegetation over the growing season [48–52], urban cover had warming effects over the growing season months and cooling effects during the rest of the year. At Ecoregion 56g, the highest seasonal and monthly warming effects, estimated at 0.05 Mg $C_{eq}$ ha$^{-1}$ gs$^{-1}$ in 2002 and 2012, and ~0.1 Mg $C_{eq}$ ha$^{-1}$ mo$^{-1}$ in June and July, are equivalent to ~29% and 45%, respectively, of the annual net ecosystem production (NEP) of deciduous forest stands in northern Michigan [53]. On the other hand, the maximum monthly cooling effect of urban landscapes at Ecoregion 56h was equivalent to $-0.7$ Mg $C_{eq}$ ha$^{-1}$ mo$^{-1}$ (in February). In comparison, Xu et al. [38] estimated a $C_{eq}$ offset between $-2.2$ Mg $C_{eq}$ ha$^{-1}$ and $-4.4$ Mg $C_{eq}$ ha$^{-1}$ induced by an increase in pavement albedo of 0.01 (i.e., 1%) across two major US cities over a 50-year period. Such results are important, considering that the watershed includes two major urban centers, Kalamazoo and Battle Creek, with a total population of >500,000 people in 2010 [36]. More so, as previous studies predicted [54], Michigan urban areas is anticipated to increase >50% by 2030.

### 4.3. Intra- and Inter-Annual Variability of Albedo and $GWI_{\Delta\alpha}$

Despite our expectation that the intra- and inter-annual variability of surface albedo would vary due to seasonality and climatic conditions [11], we did not find significant differences in inter-annual variation of $\Delta\alpha$ nor $GWI_{\Delta\alpha}$. For example, each cover type showed unique inter-annual trends that appeared to be similar during the seasonal and monthly periods. However, we found that during the growing season, the $GWI_{\Delta\alpha}$ of cropland and urban cover types was not the same in every ecoregion, which emphasizes that cover types may have different contributions depending on the location. In other words, changes in landscape composition in the five ecoregions could cause different net landscape $GWI_{\Delta\alpha}$. For example, contrasting landscape compositions among the five ecoregions led to different cumulative cooling effects during the growing season over 19 years. This value varied from $-6.89$ Mg $C_{eq}$ ha$^{-1}$ gs$^{-1}$ to $-11.36$ Mg $C_{eq}$ ha$^{-1}$ gs$^{-1}$ at Ecoregions 56d and 56b, respectively. At monthly scales, landscape composition produced cumulative cooling effects between $-9.14$ Mg $C_{eq}$ ha$^{-1}$ mo$^{-1}$ and $-14.89$ Mg $C_{eq}$ ha$^{-1}$ mo$^{-1}$ at Ecoregions 56h and 56b, respectively.

Our results also suggest that a total forest loss of ~680 ha due to the conversion to cropland and urban (i.e., the main cover type classes within the watershed) during 2001−2016 led to seasonal and monthly cooling effects at the watershed scale that, on average, were equivalent to ~$-179$ Mg $C_{eq}$ gs$^{-1}$ and ~$-374$ Mg $C_{eq}$ mo$^{-1}$, respectively.

These amounts equal approximately 90 ha and 190 ha, respectively, of mature forest net carbon sequestration in the same region, assuming an average value of 2 Mg C ha$^{-1}$ yr$^{-1}$.

### 4.4. Seasonal Percent Contributions to the Total Cooling and Warming

In line with other studies [55], the largest contribution to the overall total seasonal GWI$_{\Delta\alpha}$ came from the non-growing season months, during which all the cover types exhibited cooling effects that varied in magnitude depending on the ecoregion. Once again, urban was the only cover type that contributed to warming effects in the growing season, generally following a decreasing trend going from June to September. Such results echo the need reported in previous work to advance research on the importance of surface albedo modification within urban components (e.g., pavements, roofs, walls) as climate regulation strategy to resolve the urban energy budget and energy demand [38,56]. However, it should be noted that urban areas are composed of infrastructure (e.g., roofs, walls, pavements, etc.) and other cover types (e.g., trees, grasses, bare soil, water bodies, etc.) with varying albedo contributions, which were not considered in this study (see next section for more details). Nevertheless, the seasonal analysis clearly confirmed that the contributions to the total landscape cooling or warming effect varied by ecoregion. We found that albedo climate benefits either contributed net cooling/warming or a net neutral effect of the at landscape scale depending on the ecoregion.

### 4.5. Assumptions and Limitations of the Study

Several assumptions and limitations in our study could benefit computations of albedo-induced global warming impact (GWI$_{\Delta\alpha}$) elsewhere. The first assumption is related to the choice of the time horizon (TH, see Equation (4)) fixed at a 100-year period. The choice of either short or long time horizons can either over- or de-accentuate GWI$_{\Delta\alpha}$ values [57]. Specifically, by keeping TH fixed at 100 years, we assume that the land cover composition of the study area will remain the same for the next 100 years, although it is likely that the land cover over the next 100 years will be very different. However, by setting TH = 100, we aligned our study with the Kyoto Protocol [11,44], and hence with the IPCC protocols.

There is also uncertainty associated with the datasets employed in this study. For example, the MODIS MCD43 albedo product has a pixel with a nominal spatial resolution of 500 × 500 m, which has been shown not to properly match the effective spatial resolution (usually much higher than the nominal one [58,59]). However, previous attempts by researchers to analyze the effective spatial resolution of the MODIS albedo product [60,61] were limited to a single homogeneous area. For areas characterized by substantial land surface heterogeneity, similar to the one presented in this work, the effective spatial representativeness of the pixel is hard to determine [58]. Regarding land cover classification assumptions, we acknowledge that NLCD data are not annually available like albedo data. To overcome this limitation, when NLCD data were not available for a particular year, we assumed land cover was similar to the previous year available. Although the inter-annual LULCC for those years may not be captured, in this way, we were able to carry on a longer timeseries analysis of albedo which otherwise would have been limited to only 7 years (i.e., 2001, 2004, 2006, 2008, 2011, 2013, and 2016). A second assumption regards our analysis of urban areas. The NLCD provides four sub-classes of impervious surfaces (i.e., <20%, between 20% and 49%, between 50% and 79%, and between 80% and 100% of the total cover) composing the developed (i.e., urban) class. However, in this study, we aggregated the four sub-classes into one single "urban" land cover class. We are aware that by doing so, we may have obtained less than precise values of albedo change in urban areas. Other studies [62] have demonstrated that urban heat island intensity (UHII) is highly sensitive to the spatial context of urban areas (i.e., urban, rural, and their combination). Despite this, the focus of our study was not to investigate the importance of urban albedo modifications in the context of UHII effects. We acknowledge that a clear definition of urban and rural contexts would benefit future investigations of albedo changes in urban areas.

We investigated the contribution of landscape composition due to land transformation on $GWI_{\Delta\alpha}$ in the context of climate change mitigations by considering the forest cover type as the reference for the entire study area [24]. However, without other references, our calculations cannot estimate the forests' contribution to $GWI_{\Delta\alpha}$. This hinders a comprehensive synthesis of the total climate emission or mitigation of the watershed, considering that the low albedo of forests contributes to climate warming [63]. Moreover, regarding indirect biophysical effects, forests' role in mitigating climate change is multifaceted. Recent studies have demonstrated how re-/afforestation strategies can increase low-level cloud cover formation, which, depending on the forest type, results in cooling effects [64]. Analyzing land transformation with reference to forests is only one method. For example, other studies and policymakers have focused on land transformation in the context of bioenergy conversions [55] and land management practices to compare landscape dynamics to agriculture [65–67].

We also acknowledge that the instantaneous albedo values at 10:30 a.m. local time are likely different from the daily averages. In this study, we utilized albedo estimated by the MODIS BRDF function, which is the composite of a 16-day period [68], with albedo values from a single snapshot at 10:30 a.m. local time (e.g., MODIS Terra morning overpassing time). However, there is increasing evidence showing diurnal variations of albedo [68,69] under different sky conditions. Nevertheless, the MODIS MCD43 albedo product is soundly validated [70–72], as well as widely accepted for retrieving albedo from other remote sensing products [32,73–77] and presenting overall good accuracy compared to in situ daily averages [70].

Lastly, we considered the growing season and monthly albedo at 10:30 a.m. local time by computing the median composite (see Appendix A.1), which prevented us from accounting for the effects of land surface characteristics (i.e., vegetation properties, such as leaf area index, and landscape heterogeneity) on the spatiotemporal variation of albedo among and within patches of the same type. This represents a limitation, as during the growing season, vegetation cover and canopy structure and albedo are negatively correlated due to the varying capacity of the canopy to absorb incoming solar radiation [78]. Our cover type categories did not reflect these differences, so future efforts will be needed to quantify such differences, including the use of other remote sensing metrics and instantaneous measurements [79].

## 5. Conclusions

Albedo-induced global warming impact ($GWI_{\Delta\alpha}$) accounted for significant climate benefits (i.e., $C_{eq}$ mitigation) due to land cover changes and structural variations across the landscape. Looking at individual cover types, the climate benefits were higher in cropland, with seasonal and monthly offsets of 18% and 83%, respectively, of the annual greenhouse gas emissions of maize fields in the same area. The second-highest benefits were found in pasture lands. However, urban showed near-neutral albedo climate benefits. Notably, the overall change in landscape composition within the five ecoregions caused different net landscape $C_{eq}$ mitigations. Seasonal climate benefits ranged from $-6.89$ Mg $C_{eq}$ ha$^{-1}$ gs$^{-1}$ at Ecoregion 56d to $-11.36$ Mg $C_{eq}$ ha$^{-1}$ gs$^{-1}$ at Ecoregion 56b, and monthly climate benefits ranged from $-9.14$ Mg $C_{eq}$ ha$^{-1}$ mo$^{-1}$ at Ecoregion 56h to $-14.89$ Mg $C_{eq}$ ha$^{-1}$ mo$^{-1}$ at Ecoregion 56b. Changes in albedo due to land cover changes and landscape composition are of fundamental importance in the context of landscape climate regulation. We must couple these estimates with biogeochemical (i.e., GHGs) climate benefits to increase our understanding of the magnitude that various human-induced mechanisms contribute to climate change.

**Supplementary Materials:** The following supporting information can be downloaded at: https://www.mdpi.com/article/10.3390/land11020283/s1, Table S1: Downscaling model; Table S2: Seasonal and monthly albedo; Tables S3.1—3.5: Growing season albedo-induced GWI at ecoregions; Tables S4.1—4.5: Monthly albedo-induced GWI at ecoregions; Table S5: ANOVA model; Table S6: GWI contributions; Table S7: GWI contributions higher than average; Tables S8.1—8.5: Growing season albedo-induced RF at ecoregions; Tables S9.1—9.5; Monthly albedo-induced RF at ecoregions; Figure S1: Albedo-induced RF.

**Author Contributions:** Conceptualization, P.S. and J.C.; Data curation, P.S. and V.G.; Formal analysis, P.S., V.G., M.A. and C.L.; Funding acquisition, J.C. and G.P.R.; Investigation, J.C.; Methodology, P.S., V.G., M.A., C.L., G.S. and J.Y.; Supervision, J.C., M.A. and G.P.R.; Writing—original draft, P.S.; Writing—review & editing, P.S., J.C., V.G., M.A., C.L., G.S., J.Y. and G.P.R. All authors have read and agreed to the published version of the manuscript.

**Funding:** This study was supported by the NASA Carbon Cycle & Ecosystems program (NNX17AE16G), the Great Lakes Bioenergy Research Center funded by the U.S. Department of Energy, Office of Science, Office of Biological (DE-SC0018409) and Environmental Research (DE-FC02-07ER64494), and the Natural Science Foundation Long-term Ecological Research Program (DEB 1637653) at the Kellogg Biological Station.

**Institutional Review Board Statement:** Not applicable.

**Informed Consent Statement:** Not applicable.

**Data Availability Statement:** The data presented in this study are available on request from the author.

**Acknowledgments:** We thank Maowei Liang for helpful suggestions with statistical analyses, and Kristine Blakeslee and Jane Schuette for helpful edits and comments. The NASA POWER datasets were obtained from the NASA Langley Research Center POWER Project funded through the NASA Earth Science Directorate Applied Science Program. We also thank the two anonymous reviewers who helped to improve the quality of our manuscript.

**Conflicts of Interest:** The authors declare no conflict of interest.

## Appendix A. Materials and Methods

*Appendix A.1. Google Earth Engine Processing*

Instantaneous albedo data at 10:30 a.m. local time (MODIS Terra morning overpassing time) from the MODIS Bidirectional Reflectance Distribution Function (BRDF) MCD43A3 (v006) product [31] was produced by the inversion of a BRDF model against a 16-day moving window of MODIS observations at $500 \times 500$ m spatial resolution. For each image, we selected the "Albedo_WSA_shortwave" (i.e., white-sky albedo) band and rescaled it to 0–1. For quality control, we applied the quality band "BRDF_Albedo_Band_Mandatory_Quality_shortwave" (i.e., the full BRDF inversion) [11,80] by filtering out pixels not meeting the control protocols. We then used an additional quality band ("Snow_BRDF_Albedo") from the MCD43A2 product [81] to further filter and select quality snow-albedo retrievals in the MCD43A3 product. To obtain the growing season at watershed level for each year, we applied the same methodology as Jeong et al. [33] and Sciusco et al. [11]. Briefly, we used the enhanced vegetation index (EVI) to identify the growing season by detecting the EVI inflection points (i.e., the dates) when maximum and minimum change rate in greenness occurred over the entire watershed. We obtained 16-day composite time series of EVI at a $250 \times 250$ m spatial resolution from the most recent collection (V006) of the MODIS MYD13Q1 product [82]. Similar to the quality control protocols for albedo product, we filtered and selected only good quality EVI pixels by applying the quality band "SummaryQA" from the MYD13Q1 product. Both albedo and EVI acquisitions referred to approximately 10:30 a.m. and 1:00 p.m. local time, respectively, when MODIS (Terra and Aqua, respectively) passes over the study area. Lastly, we created median composite of the albedo data into growing season (i.e., roughly from March—November for the 19-year period) and monthly (i.e., January–December, over the 19-year period, less March, which we removed because few images were available and likely due to high cloud cover) time-steps at 10:30 a.m. local time.

*Appendix A.2. Clearness Index ($K_T$) for Calculation of Albedo-Induced Radiative Forcing at the Top-of-Atmosphere ($RF_{\Delta\alpha}$)*

Instantaneous albedo-induced radiative forcing at the top-of-atmosphere ($RF_{\Delta\alpha}$; see Equation (3)) using the solar radiation at the Earth's surface is calculated [37,38] as:

$$RF_{\Delta\alpha} = -(SW_{TOA} \cdot \Delta\alpha_p) \tag{A1}$$

where $SW_{TOA}$ is the incident shortwave radiation at the top-of-atmosphere and $\Delta\alpha_p$ is the change in planetary albedo. Changes in planetary albedo ($\Delta\alpha_p$) are linearly related to changes in surface albedo ($\Delta\alpha_s$) as follows [37,42,83,84]:

$$\Delta\alpha_p = f_a \cdot \Delta\alpha_s \tag{A2}$$

where $f_a$ is a two-way atmospheric transmittance parameter that accounts for both the reflection and absorption of solar radiation through the atmosphere, and it can be decomposed into downward and upward transmittance coefficients as follows [83]:

$$f_a = K_T \cdot T_a \tag{A3}$$

where the clearness index $K_T$ is the fraction of $SW_{TOA}$ reaching the Earth's surface, $T_a$ is the upward atmospheric transmittance factor (i.e., the fraction of the radiation reflected by the Earth's surface back at the top-of-atmosphere). In turn, $T_a$ is calculated as:

$$T_a = \frac{SW_{in}}{SW_{TOA}} \tag{A4}$$

where $SW_{in}$ and $SW_{TOA}$ are the incident shortwave radiation at the surface and at the top-of-atmosphere, respectively. By replacing Equations (A2)–(A4) in (A1), we obtain Equation (3), which is reproduced here for the reader's convenience:

$$RF_{\Delta\alpha} = -(SW_{in} \cdot K_T \cdot \Delta\alpha_s) \tag{A5}$$

where $RF_{\Delta\alpha}$ (W m$^{-2}$) is the instantaneous albedo-induced radiative forcing at the top-of-atmosphere at 10:30 a.m. local time, $SW_{in}$, $K_T$, and $\Delta\alpha_{si}$ are the incident shortwave radiation at the surface, the clearness index, and the surface albedo difference between a cover type *i* and the reference forest for growing season and monthly periods and for each ecoregion (see Equation (2)).

We also calculated $RF_{\Delta\alpha}$ by using the upward atmospheric transmittance factor ($T_a$; Equation (A4)) as in Carrer at al. [13] and Sciusco et al. [11], although the differences in $RF_{\Delta\alpha}$ calculated with the use of $K_T$ and $T_a$ were negligible, so we decided to only report $RF_{\Delta\alpha}$ calculated with $K_T$.

*Appendix A.3. Carbon-Dioxide Airborne Fraction (AF(t))*

The carbon-dioxide ($CO_2$) airborne fraction that remains in the atmosphere at time (t) following a single pulse emission is modeled with an exponential function through multi-model impulse response function analysis [40] as follows:

$$AF(t) = a_0 + \sum_{i=1}^{3}\left[a_i e^{\frac{-t}{\tau_i}}\right] \tag{A6}$$

where *t* represents the time in years, $a_i$ and $\tau_i$ are the fitted coefficients representing the decay of $CO_2$ pulse emission in the atmosphere over time. The recommended mean coefficients obtained from the multi-model impulse response function analysis are: $a_0 = 0.2173$ (the fraction of $CO_2$ that remains permanently in the atmosphere); $a_1 = 0.2240$; $a_2 = 0.2824$; $a_3 = 0.2763$; $\tau_1 = 394.4$; $\tau_2 = 36.54$; and $\tau_3 = 4.304$ [40].

*Appendix A.4. Analysis of Variance (ANOVA)*

Prior to running the analysis of variance (ANOVA) (Equations (5) and (6)), we checked for normal distribution of the residuals (i.e., normality and heteroscedasticity assumptions) and outliers, and performed the Mauchly's test (i.e., sphericity assumption) when necessary. Wherever the sphericity assumption was violated, we applied the Greenhouse-Geisser corrections. We then calculated the generalized eta-squared ($\eta^2$) [85] to examine the variance of dependent variable $GWI_{\Delta\alpha}$ by ecoregion, cover type and their interactions. Lastly, we carried out a post-hoc Tukey test analysis to see whether differences in the least square means (LSMs) of $GWI_{\Delta\alpha}$ among and within cover type and ecoregions were significant. All analyses were carried out in RStudio v. 1.2.5033 [35], using the R-packages "*ez*", "*nlme*", "*lsmeans*", and "*multcomp*" [86–89].

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
