# Peer review of "Albedo-Induced Global Warming Impact at Multiple Temporal Scales within an Upper Midwest USA Watershed"

_land, doi:10.3390/land11020283_

Round 1

Reviewer 1 Report

This paper is very attractive to assess the albedo change along with land use land cover changes. Authors also present a clear introduction to tell people how the albedo change contributes to climate change with a metric of carbon equivalent. However, I believe should do some (major) revisions before its next-round process.
1. Line 22, please remove the contents on 2-3 thousand years.
2. same for line 48. 
3. line 59, typo, should be LULCC
4. line 109, today? in which year? you need to consider it won't be 'today' for future readers.
5. Have you considered the variation of albedo in a day? moreover, the albedo also indicates seasonal or month variations. Please discuss the inaccuracy caused by the diurnal, monthly or seasonal variations of albedo in the discussion or limitation section.
6. I believe it could be more meaningful to present the specific months rather than growing months. Growing months include different growing stages of plants. A table is needed to present them or you need to present them in the appendix.
7. Line 145-146, a brief description of the methods is needed.
8. It might be problematic to show the urban as urban areas can be further divided into different land cover types such as the built-up types and land cover (soil, water, vegetation types). They also indicate different albedo(s) and associated land surface temperature properties. See paper: Spatial Variability and Temporal Heterogeneity of Surface Urban Heat Island Patterns and the Suitability of Local Climate Zones for Land Surface Temperature Characterization. Remote Sensing, 13(21), 4338.
9. A subsection of limitation should be presented before conclusions
10. line 426, land use land cover changes here is too vague. Please include how the land use land cover changes, corresponding to cooling potential. it will be more attractive if you can further explain the land cover type, ecoregion....
11. Research significance should be presented at the end of the conclusions.

Reviewer 2 Report

The manuscript studies changes in landscape albedo-induced at multiple temporal scales, specifically seasonal and monthly, over a 19-year period for different land cover types in five ecoregions. This is an interesting topic for readers and appropriate for the journal and is generally well written.

Comments:

Lines 123-127: I suggest that the authors add references to support the use of this method.

Lines 129-131: The map projection should be added.

Figure 2 (lines 251-256) refers to table B3 of the supplementary materials with the values that have been represented in that figure. For ease of reading it would be advisable to simply add the values to the figure. The same for lines 257-263 (figure 3 and table B4).

Lines 347-348. The phrase "substantial influence of snow cover on land surface albedo" is not very specific. One could, for example, take into account that albedo depends on grain size, snow density and the snow age.

As for the supplementary materials, in addition to the previous recommendations (tables B3 and B4), I suggest moving to section 2.2 of the manuscript the paragraph from line 119-126, which appears as part of the “Assumptions and limitations of the study”. In addition, it would be of interest if the authors could expand on the references of studies that support the use of MODIS, specifically the MCD43 product.

References:

Wang, Z. S.; Schaaf, C. B.; Chopping, M. J.; Strahler, A. H.; Wang, J.; Román, M. O.; Rocha, A. V.; Woodcock, C. E.; Shuai, Y. M. Evaluation of Moderate-resolution Imaging Spectroradiometer (MODIS) snow albedo product (MCD43A) over tundra. Remote Sens. Environ. 2012, 117, 264–280, doi:10.1016/j.rse.2011.10.002.

Qu, Y.; Liu, Q.; Liang, S.; Wang, L.; Liu, N.; Liu, S. Direct-Estimation Algorithm for Mapping Daily Land-Surface Broadband Albedo from MODIS Data. IEEE Trans. Geosci. Remote Sens. 2014, 52, 907–919.

Corbea-Pérez, A.; Calleja, J.F.; Recondo, C.; Fernández, S. Evaluation of the MODIS (C6) Daily Albedo Products for Livingston Island, Antarctic. Remote Sens. 2021, 13, 2357, doi:10.3390/rs13122357.

Round 2

Reviewer 1 Report

The authors seriously responded the comments and have made proper modification to the manuscript. i suggest the authors take a couple of minutes to look through the text again and have a another check for spellings. submit the manuscript after final minor revision.
